

# Morphology and cytochemical patterns of peripheral blood cells of tiger frog (*Rana rugulosa*)

Xianxian Chen, Yu Wu, Lixin Huang, Xue Cao, Misbah Hanif, Fei Peng, Xiaobing Wu and Shengzhou Zhang

College of Life Sciences, Anhui Normal University, Wuhu, Anhui Province, China

Corresponding author
Shengzhou Zhang,
szzhang@mail.ahnu.edu.cn

## ABSTRACT

**Background**. Tiger frog (*Rana rugulosa*) is a national second-class protected amphibian species in China with an important ecological and economic value. In recent years, due to excessive human hunting, pollution and habitat loss, the wild population of tiger frog has declined sharply. To protect wildlife resources, the artificial breeding of tiger frogs has rapidly developed in China. Diseases are increasing and spreading among tiger frogs due to the increasing scale of artificial farming. The blood examination is the most straightforward and less invasive technique to evaluate the animal health condition. Thus, it is essential to obtain the normal hematological indicators of tiger frogs. The objective of this study was to investigate the morphometry, microstructure and cytochemical patterns of peripheral blood cells in tiger frogs.

**Methods**. The number of blood cells in tiger frogs was counted on a blood count board, and the cell sizes were measured by a micrometer under light microscope. The morphology and classification of blood cells were studied by Wright-Giemsa staining, and the cytochemical pateerns was investigated by various cytochemical staining including periodic acid-Schiff (PAS), Sudan black B (SBB), peroxidase (POX), alkaline phosphatase (AKP), acid phosphatase (ACP), chloroacetic acid AS-D naphthol esterase (CAE) and $\alpha$-naphthol acetate esterase (ANAE) staining.

**Results**. Besides erythrocytes and thrombocytes, five types of leukocytes were identified in tiger frogs: neutrophils, eosinophils, basophils, lymphocytes and monocytes. The mean erythrocyte, leukocyte and thrombocyte counts were $1.33 \pm 0.15$ million/mm$^3$, $3.73 \pm 0.04 \times 10^4$/mm$^3$ and $1.7 \pm 0.01 \times 10^4$/mm$^3$, respectively. Small lymphocytes were the most abundant leukocytes, followed by large lymphocytes, Neutrophils, eosinophils and monocytes, basophils were the fewest. Eosinophils were strongly positive for PAS, positive for SBB, POX, ACP, CAE, ANAE, while weakly positive for AKP staining; basophils were strongly positive for PAS, ACP, positive for SBB, CAE, weakly positive for ANAE, negative for AKP, POX staining; neutrophils were strongly positive for ACP, SBB, positive for PAS, POX, weakly positive for AKP, CAE and ANAE staining; monocytes were positive for PAS, SBB, ANAE, weakly positive for ACP, AKP, POX, CAE staining; large lymphocytes and thrombocytes were positive for PAS, ACP, weakly positive for ANAE, while negative for SBB, POX, AKP, CAE; small lymphocytes were similar to large lymphocytes, except for strongly positive for PAS and ACP staining.

**Conclusions**. The blood cell types and morphology of tiger frogs were generally similar to those of other amphibians, while their cytochemical patterns had some notable species specificity.Our study could enrich the knowledge of peripheral blood cell

morphology and cytochemistry in amphibians, and provide baseline data for health condition evaluation and disease diagnosis of tiger frogs.

# INTRODUCTION

Blood cells are vital to the animal body. The vertebrate blood cells can be grouped into erythrocytes, leukocytes and thrombocytes. The erythrocytes serve as carriers for transporting oxygen and carbon dioxide (*Fang et al., 2014*; *Yoshida, Prudent & Alessandro, 2019*). The leukocytes are responsible to protect the body against both infectious disease and foreign invaders, while the thrombocytes play a major role in hemostasis and coagulation (*Salakij et al., 2000*; *Arikan & Cicek, 2014*; *Peng et al., 2018*). Blood cells are sensitive to changes in the animal internal physiological states and stimuli from the external environment, the morphology and number of different types of blood cells could reflect the health status of the animals, and their abnormal variations may be associated with inflammation, pathogenic microorganism infection or other diseases (*Fang et al., 2014*; *Peng et al., 2018*; *Kehoe et al., 2020*). Therefore, the normal hematological data can be used for animal health monitoring.

Wright-Giemsa staining was widely used to observe the cell morphology and identify the blood cell types; meanwhile, various cytochemical stains were employed to detect the cellular chemical composition and recognize the functions of different blood cell types (*Tavares-Dias & Marques-Barcellos, 2005*; *Kehoe et al., 2020*; *Oliveira et al., 2021*). Intracellular glycogen and lipid can be displayed by periodic acid-Schiff (PAS) and Sudan black B (SBB) staining, respectively, which may provide energy for phagocytosis (*Ueda et al., 2001*). Gomori lead sulfide and kaplow azo coupling reaction could display acid phosphatase (ACP) and alkaline phosphatase (AKP) in the blood cells, respectively. ACP and AKP are lysosomal enzymes also related to the phagocytic process, being discharged into the phagocytic vacuoles of leukocytes that have ingested bacteria and other particles (*Hirsch & Cohn, 1964*). Tetramethylbenzidine reaction could display peroxidase enzyme (POX) in the blood cells. POX is a lysosomal enzyme which plays a significant role in the defense against bacterial infection (*Bielek, 1981*; *Dvorak, Estrella & Ishizaka, 1994*). Chloroacetic acid AS-D naphthol esterase (CAE) is a specific granulocyte esterase, which may be responsible for cellular defense, facilitating diapedesis, toxic product and microorganism inactivation, while $\alpha$-naphthol acetate esterase (ANAE) is a non-specific esterases, which may play a crucial role in the processing and antigen-presenting of intracellular toxin and small molecules (*Ueda et al., 2001*; *Azevedo & Lunardi, 2003*).

At present, the classification, morphology and cytochemistry of peripheral blood cells have been generally investigated in various vertebrates, especially in human and mammals (*Salakij et al., 2000*; *Techangamsuwan et al., 2010*; *Fang et al., 2014*; *Hernández et al., 2017*). Amphibians are the transitional group of vertebrates from aquatic to terrestrial mode

of life. In recent decades, a pandemic loss in amphibian biodiversity has occurred due to the destruction of ecological environment, climate change, and the spread of diseases (*Zhou et al., 2004*; *Bricker, Raskin & Densmore, 2012*; *Isaak-Delgado et al., 2020*; *Gavel et al., 2021*). However, the immune system and the defence mechanisms of amphibians are poorly known, relatively few studies were reported on the morphology and especially the cytochemistry of amphibian blood cells. *Madhusmita & Kumari (2014)* have been only described the morphological features of dubois's tree frog (*Polypedates teraiensis*) peripheral blood cells; *Gutierre et al. (2008)* have been only detected the glycogen particles, lipids and peroxidase in the caecilian *Siphonops annulatus* granulocytes; *Bricker, Raskin & Densmore (2012)* compared the morphology and four cytochemical stains in the peripheral blood cells of American bullfrog (*Rana catesbeiana*) and African clawed frog (*Xenopus laevis*), finding erythrocytes and leukocytes of American bullfrog were larger than those of African clawed frog, as well as significant species specific differences in cell percentage and cytochemical patterns of the blood cells between these two amphibian species.

Tiger frog (*Rana rugulosa*), belonging to Amphibia, the Ranidae of Anura, is a species of large edible frog widely distributed from southwest and south of china to south and southeast of Asia (*Tian, Xia & Shao, 2011*). The excellent flesh quality and high nutritional values make this species popular with consumers in China. In recent 20 years, due to excessive human hunting, pollution and habitat loss, the wild population of tiger frog has declined sharply in various places, and even become endangered in some areas (*Tian, Xia & Shao, 2011*; *Li et al., 2014*). To protect wildlife resources and meet the market demand, tiger frog has been listed as a national second-class protected species, and its artificial breeding has rapidly developed in China. However, diseases are increasing and spreading among tiger frogs due to the increasing scale of artificial farming. Routine monitoring frog health is crucial to reduce the incidence of diseases. Haematological traits and parameters are significant indicators of the physiological and health status of animals (*Xiong et al., 2018*; *Ballester et al., 2020*). The blood examination is the most straightforward and less invasive technique to evaluate the animal health condition. Thus, it is essential to obtain the normal hematological indicators of tiger frogs. At present, the basic research on tiger frog has been extensively carried out (*Xie et al., 2009*; *Tian, Xia & Shao, 2011*; *Li et al., 2014*); however, until now, little information is available concerning the morphology and especially the cytochemistry of its peripheral blood cells. In this study, we investigated the classification, morphology, counts, the percentage and cytochemical patterns of the tiger frog blood cells by Wright-Giemsa staining and a range of cytochemical staining techniques. Our results could enrich the knowledge of peripheral blood cells in frogs, and provide reference data for the artificial culture and health examination of tiger frogs.

## MATERIALS & METHODS

### Animals and blood smears preparation

Fifteen male and fifteen female adult healthy tiger frogs (mean weight: 0.25–0.50 kg) were obtained from local frog farms in Wuhu City from June to September 2018; All frogs have normal appearance without obvious signs of disease. This work was approved by the ethics committee of Anhui Normal University (Approval No. 201811).
Approximately 0.3 ml blood was taken from the ventral abdominal vein using a sterile 1 ml syringe with 26g needles and placed quickly into a K$_2$-EDTA anticoagulant blood collection tube. Blood smears were then immediately prepared, air-dried, and stored at 4 °C for further Wright-Giemsa and various cytochemical staining.

### Wright-Giemsa staining

Following methanol fixation for 15 min, the blood smears were treated with Wright-Giemsa reagent according to the guidelines specified by the Biological Engineering Co. Ltd. (Shanghai, China). Briefly, blood smears were stained with Wright-Giemsa reagent for 1 min at room temperature, then placed in a phosphate buffer(pH7.2) for 15 min. Rinsing with running water to prevent sediment settling on the blood smears. The stained blood smears were observed and photographed with an Olympus BX61 light microscope (Olympus, Tokyo, Japan).

### Cytochemical staining

The cytochemical staining was performed according to the methods of *Fang et al. (2014)*, *Chen et al. (2018)* and *Chen et al. (2019)* with minor modifications, The prepared blood smears were fixed with 95% ethanol solution for periodic acid-schiff (PAS) reaction, formaldehyde vapor for sudan black B (SBB) staining and gomori lead sulfide (ACP) reaction, 10% methanolformaldehyde solution for kaplow azo coupling (AKP) reaction and chloroacetic acid AS-D naphthol esterase (CAE) staining, and 10% formaldehyde normal saline for $\alpha$-naphthol esterase (ANAE) staining. The specific staining procedures were briefly described below. The healthy human blood smears were used as controls for cytochemical staining to ensure all staining worked as expected.

### Periodic acid-Schiff reaction for glycogen

Blood smears were oxidized by 10 mg/ml periodate solution for 15 min, and stained with the Schiff's solution for 60 min. They were then rinsed with sulfite solution for three times and distilled water for 2–3 min, successively. After dried at room temperature, they were counterstained with 20 mg/ml methyl green solution for 15–20 min.

### Sudan black (SBB) staining for lipids

Blood smears were placed in sudan black B solution (300 mg SBB dissolved in 100 ml 70% ethanol) at 37 °C for 60 min. They were then rinsed with 70% ethanol for 1–2 min and double distilled water for 1 min, successively. After dried at room temperature, they were counterstained with Wright-Giemsa reagent for 15–20 min.

### Gomori lead sulfide reaction for ACP

Blood smears were placed in the incubation for 4 h at 37 °C, which was composed of 12 ml pH 4.7 acetic acid buffer, 2 ml 20 mg/ml lead nitrate, 4 ml 32 mg/ml $\beta$-glycerin sodium phosphate and 74 ml double distilled water. They were then washed with distilled water for 5min and stained with 10% ammonium sulfide solution for 30 min.

### Kaplow azo coupling reaction for AKP

Blood smears were stained with the matrix incubation solution, which was prepared with 20 mg alpha-phosphate naphthol sodium, 20 ml of 0.05 mol/L propylene glycol buffer and

20 mg diazo fast blue, for 45–60 min at room temperature. They were washed with double distilled water for 2 min and dried at room temperature, then they were counterstained with hematoxylin for 5–8 min.

### Tetramethylbenzidine reaction for POX

Each blood smear was stained with benzidine solution, which was mixed by 3 ml of 0.1% tetramethylbenzidine ethanol solution and 30 $\mu$l of nitrosyl ferricyanide saturated solution, for 1 min at room temperature. Then, they were oxidized by 0.7 ml of 1% $H_2O_2$ for 6 min. Rinsed with tap water directly and dried at room temperature. Smears were counterstained with Wright-Giemsa reagent for 15–20 min.

### Chloroacetic acid AS-D naphthol staining for CAE

Blood smears were placed in the staining solution, which was perpared by dissolving solution A(10 mg chloroacetic AS-D naphthol dissolved in 0.5 ml acetone) and solution B (10 mg of diazo fast blue dissolved in 5 ml double distilled water) in 5 ml veronal acetate buffer, for 30–40 min at 37 °C. After being washed with double distilled water and dried at room temperature, the smears were counterstained with 1 mg/ml hematoxylin for 5–10 min.

### $\alpha$-naphthol acetate staining for ANAE

Blood smears were stained with the incubation solution, which was prepared by solution A(400 mg of $\alpha$-naphthol acetate dissolved in 2 ml of 50% aceton) and solution B(100 mg diazo fast blue dissolved in 100 ml of 0.067 mol/L phosphate buffer) at 37 °C for 1 h. They were washed with double distilled water and dried at room temperature, then the smears were counterstained with 10 g/L methyl green solution for 5–15 min.

### Evaluation of cytochemical staining intensity

With regard to cytochemical activity, the results of cytochemical staining were divided into four reaction types based on the evaluation methods described by *Tavares-Dias & Marques-Barcellos (2005)*: negative reaction (−), weakly positive reaction (+), positive reaction (++) and strongly positive (+++).

### Blood cell counts and measurements

The total erythrocyte, leukocyte and thrombocyte counts were determined manually with the Neubauer chamber as described previously (*Wang et al., 2021*). The number of erythrocyte, leukocyte and thrombocyte were calculated according to the proportions of these cells counted on the wright's blood smears. The percentages of different leukocyte types were calculated after counting 3,000 randomly selected leukocytes from male and female individuals. The cell sizes were manually measured using ocular micrometer. Twenty of each type of blood cells were randomly selected from each frog to measure size and 100 leukocytes were randomly selected from each frog to calculate the percentage of various leukocytes.

### Statistical analysis

Hematology data and morphometric values were presented as means and standard deviation (SD). Statistical comparisons of the differences in erythrocyte, leukocyte and

thrombocyte counts were performed using Poisson distribution GLM with R revision 3.6.1 (*R Core Team, 2019*), and comparisons of percentage in different leukocytes between sexes were performed using quasibinomial distribution GLM model analysis. In addition, the morphometric values among different cell types were performed using independent sample T test analysis (SPSS 19.0, SPSS Inc., Chicago, IL, USA). A *p*-value less than 0.05 was considered significant.

# RESULTS

In this study, erythrocytes, thrombocytes and five types of leukocytes: neutrophils, eosinophils, basophils, monocytes and lymphocytes were distinguished in the peripheral blood cells of tiger frog under light microscopy by Wright-Giemsa staining. The lymphocytes can be subdivided into large and small lymphocytes depending on the size of the nucleus and cytoplasm.

## The blood cell counts and percentage of leukocytes

The mean erythrocyte, leukocyte and thrombocyte counts were $1.33 \pm 0.15$ million/mm$^3$, $3.73 \pm 0.04 \times 10^4$/mm$^3$ and $1.7 \pm 0.01 \times 10^4$/mm$^3$, respectively. There were no statistically significant differences in the erythrocyte, leukocyte and thrombocyte counts between female and male tiger frogs ($P > 0.05$). Small lymphocytes were the most abundant leukocytes, followed by neutrophils and large lymphocytes, eosnophils and monocytes were fewer, basophils were the fewest. There were no statistically significant differences in the total number of leukocytes, the percentage of various leukocytes and the neutrophil to lymphocyte (N/L) ratio between male and female frogs (Table 1).

## The size and morphology of tiger frog peripheral blood cells
### Erythrocytes

The mature erythrocytes were oval and elliptical in shape with a large central, round or long elliptical, purple-stained nucleus and an abundant, dark red brown-stained cytoplasm free of granules (Fig. 1A). Immature erythrocytes were typically smaller and rounder than mature ones with a round and relatively larger nucleus (Figs. 1B and 1C), some of which appeared hypochromatic and polychromatic (Fig. 1C). The dividing erythrocytes were sporadically found in the peripheral blood of tiger frogs (Fig. 1D). No statistically significant difference in the size of erythrocytes was found between male and female tiger frogs, while the erythrocyte nucleus of males was significantly larger than that of females ($P < 0.01$) (Table 2). There were no significant differences in the total number of erythrocytes between male and female frogs (Table 1).

### Eosinophils

Eosinophils were round or oval, frequently irregularly outlined cells. They had an eccentric, round or sometimes bilobed nucleus. The purple-stained cytoplasm was filled with coarse, round or rod-shaped, orange acidophilic granules of different size that occasionally obscured the nucleus (Fig. 1E).

**Table 1 The blood cell counts and leukocytes percentage of peripheral blood cells in tiger frogs.**

| | Females (N = 15) | | Males (N = 15) | | Model | T value | p value |
|---|---|---|---|---|---|---|---|
| | Mean ± SD | Range | Mean ± SD | Range | | | |
| Erythrocyte count ($\times 10^6$/mm$^3$) | 1.33 ± 0.15* | 1.12–1.47 | 1.12 ± 0.08 | 1.01–1.19 | Poisson | −2.217 | 0.207 |
| Leukocyte count ($\times 10^4$/mm$^3$) | 3.73 ± 0.04 | 3.19–4.12 | 3.2 ± 0.02 | 2.86–3.48 | Poisson | −0.367 | 0.714 |
| Thrombocyte count ($\times 10^4$/mm$^3$) | 1.7 ± 0.01 | 1.59–1.78 | 1.9 ± 0.04 | 1.52–2.40 | Poisson | 0.224 | 0.823 |
| Eosnophils (%) | 7.82 ± 0.67 | 7.08–8.39 | 10.21 ± 1.98 | 7.96–11.72 | Quasibinomial | 1.978 | 0.119 |
| Basophils (%) | 2.59 ± 0.42 | 2.15–2.98 | 1.86 ± 0.36 | 1.63–2.28 | Quasibinomial | −2.296 | 0.083 |
| Neutrophils (%) | 17.22 ± 1.90 | 15.97–19.40 | 20.62 ± 1.23 | 19.66–22.01 | Quasibinomial | 2.606 | 0.060 |
| Monocytes (%) | 12.55 ± 2.10 | 10.94–14.92 | 16.08 ± 1,61 | 14.88–17.91 | Quasibinomial | 2.313 | 0.082 |
| Large lymphocytes (%) | 27.85 ± 2.36 | 25.75–30.41 | 23.97 ± 2.21 | 20.5–24.75 | Quasibinomial | −2.614 | 0.059 |
| Small lymphocytes (%) | 31.69 ± 1.78 | 29.85–33.40 | 28.12 ± 4.85 | 22.8–32.30 | Quasibinomial | −1.195 | 0.298 |
| Neutrophil/lymphocyte ratio | 0.36 ± 0.04 | 0.38–0.41 | 0.41 ± 0.03 | 0.37–0.44 | Quasibinomial | 0.729 | 0.420 |

Notes.
*The difference between males and females is significant ($P < 0.05$).

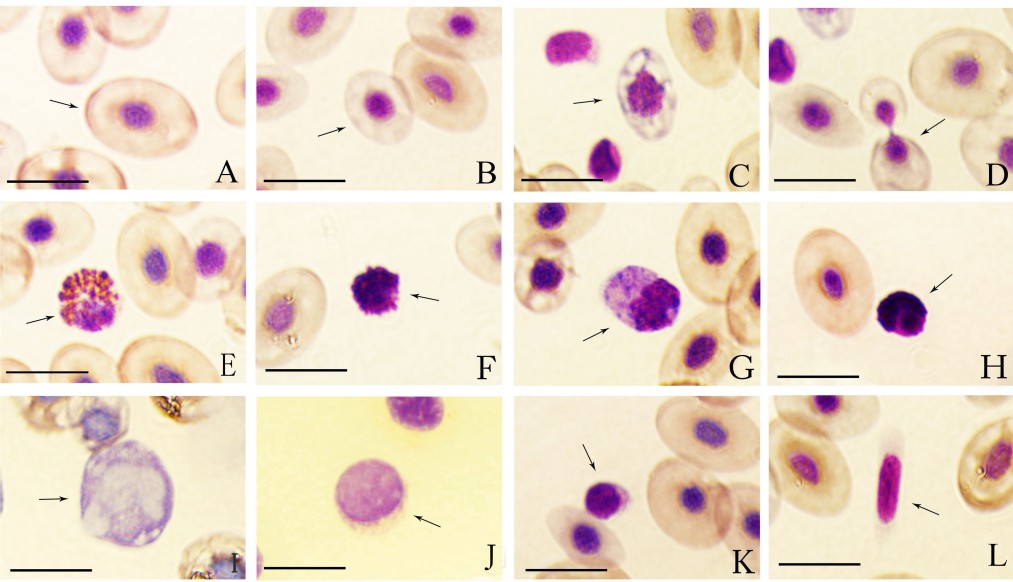

**Figure 1 Wright-Giemsa staining of tiger frog blood cells.** (A) Erythrocyte, ellipsoid shape with light red brown-stained cytoplasm free of granules; (B) immature erythrocyte; (C) immature erythrocyte with hypochromasia and polychromasia; (D) dividing erythrocyte; (E) eosinophil, round cell with an eccentric nucleus, and the cytoplasm was filled with coarse, round and orange acidophilic granules of different size; (F) basophil, round cell with a central-to-eccentric rounded and dark purple-stained nucleus; (G) neutrophil, spindle-shaped cells with numerous purple-stained granules; (H) oval neutrophil with a lobed nucleus; (I) monocyte, oval cell with a prominent, eccentric, kidney-shaped nucleus; (J) large lympho-cyte, round cell with a bigger and lightly purple-stained nucleus; (K) small lymphocyte, round cell with a smaller but dark purple-stained nucleus; (L) thrombocyte, spindle-shaped cell with an elongated spindle and lightly purple-stained nucleus; Bar = 10 μm.

**Table 2** The morphological parameters of peripheral blood cells in tiger frogs (Mean ± SD, μm, N = 20, df = 38, T test).

| Cell types | Cell length (CL) | | Cell width (CW) | | T vaule | T vaule | P vaule | P vaule |
| --- | --- | --- | --- | --- | --- | --- | --- | --- |
| | Females | Males | Females | Males | (CL) | (CW) | (CL) | (CW) |
| Erythrocytes | 15.09 ± 1.26 | 14.61 ± 1.47 | 10.60 ± 0.87 | 9.83 ± 1.10 | 0.966 | 1.771 | 0.340 | 0.085 |
| (nuclei) | (5.38 ± 0.46) | (8.56 ± 1.21)[*] | (4.42 ± 0.48) | (5.56 ± 0.37)[*] | −10.688 | −8.152 | 0.001 | 0.001 |
| Eosnophils | 13.76 ± 0.55 | 13.92 ± 1.94 | 11.26 ± 0.75 | 11.25 ± 0.76 | −0.541 | 0.052 | 0.592 | 0.959 |
| Basophils | 9.28 ± 1.00 | 8.74 ± 1.27 | 7.99 ± 0.91 | 7.69 ± 1.15 | 1.483 | 0.912 | 0.146 | 0.368 |
| Neutrophils | 15.93 ± 2.68 | 14.51 ± 2.68 | 12.59 ± 1.99 | 12.03 ± 3.32 | 1,848 | 0.646 | 0.072 | 0.522 |
| Monocytes | 22.60 ± 2.36 | 23.03 ± 2.11 | 17.70 ± 1.43 | 17.70 ± 1.68 | −0.597 | −0.068 | 0.554 | 0.547 |
| Large lymphocytes | 10.52 ± 1.25 | 10.00 ± 1.07 | 9.20 ± 1.25 | 8.79 ± 0.80 | 1.509 | 1.135 | 0.140 | 0.263 |
| Small lymphocytes | 7.18 ± 0.81 | 7.33 ± 1.08 | 6.20 ± 0.89 | 6.42 ± 0.94 | −0.501 | −0.738 | 0.619 | 0.465 |
| Thrombocytes | 17.83 ± 1.94 | 17.36 ± 1.89 | 4.23 ± 0.43 | 4.73 ± 0.52 | 0.002 | −3.331 | 0.444 | 0.456 |

Notes.
*The difference between males and females is extremely significant ($P < 0.01$).

## Basophils

Basophils were the smallest granulocyte in tiger frogs. They were difficult to find in the blood smears of tiger frogs. These cells were round with a central-to-eccentric rounded and dark purple-stained nucleus and serrated edges. The usually scarce cytoplasm contained numerous round and deep purple-stained basophilic granules, which often masked the nucleus (Fig. 1F).

## Neutrophils

Neutrophils were the largest granulocyte in tiger frogs. They were usually oval and irregular in shape. They had an eccentric, oval or round, dark violet-stained nucleus and abundant violet-stained cytoplasm with numerous blue-stained granules and several irregular blue cytoplasmic inclusions (Fig. 1G). Some neutrophils had a lobed nucleus, usually 2–5 lobes connected by the filaments (Fig. 1H).

## Monocytes

Monocytes were the largest leukocyte in tiger frogs. They were round, oval or irregular cells, characterized by a prominent, eccentric, kidney-shaped and blue purple-stained nucleus. These cells exhibit an agranular gray-blue cytoplasm, in which numerous round or oval vacuoles of different sizes were usually observed (Fig. 1I).

## Lymphocytes

Lymphocytes were round or oval in shape and had a large, purple-stained nucleus; Large and small lymphocytes were observed according to their diameters and relative amounts of cytoplasm. Large lymphocytes had a bigger and lightly purple-stained nucleus with a greater quantity of cytoplasm (Fig. 1J), while small lymphocytes had a smaller but dark purple-stained nucleus, and a rim of scant, light purple cytoplasm that was not always visible around the entire nuclear margin (Fig. 1K).

## Thrombocytes

Thrombocytes were often rod-shaped or long spindle-shaped with an elongated spindle and lightly purple-stained nucleus, and the cytoplasm was faint purple-stained with smooth

or irregular membranes. The scant cytoplasm was accumulated in the two poles when the thrombocytes were long spindle-shaped (Fig. 1L).

## The cytochemical staining features of peripheral blood cells in tiger frogs

The cytochemical staining features of peripheral blood cells of tiger frogs were shown in Fig. 2. The staining patterns of various cell types were summerized in Table 3.

### Periodic acid-Schiff reaction for glycogen

Eosinophils, basophils and small lymphocytes were strongly positive with a great number of coarse deep purple red-stained granular deposits (Figs. 2AA, 2AB and 2AF). Neutrophils, monocytes, large lymphocytes and thrombocytes (Figs. 2AC, 2AD, 2AE and 2AG) were all positive with a diffusely or granular purple red-stained cytoplasm.

### Sudan black (SBB) staining for lipids

Neutrophils presented strongly positive for SBB staining, the cytoplasm was filled with a large number of coarse and dark black-stained granules, which masked the nuclei (Fig. 2BC); eosinophils (Fig. 2BA), basophils (Fig. 2BB) and monocytes (Fig. 2BD) exhibited positive reactions with numerous dark black-stained granules; large lymphocytes (Fig. 2BE), small lymphocytes (Fig. 2BF) and thrombocytes (Fig. 2BG) were negative with reddish cytoplasm.

### Gomori lead sulfide reaction for ACP

Basophils, neutrophils and small lymphocytes were strongly positive with a great number of black brown granular or diffuse granules in the cytoplasm (Figs. 2CA, 2CB, 2CC and 2CF); eosinophils, large lymphocytes and thrombocytes were positive with numerous dark brown granular or diffuse granules in the cytoplasm (Figa. 2CE and 2CG); monocytes were weakly positive for ACP staining with numerous brownish yellow diffuse granules (Fig. 2CD).

### Kaplow azo coupling reaction for AKP

Eosinophils (Fig. 2DA), neutrophils (Fig. 2DC) and monocytes (Fig. 2DD) presented weakly positive for AKP with light gray diffuse staining in cytoplasm. Basophils (Fig. 2DB), monocytes (Fig. 2DD), large lymphocytes (Fig. 2DE), small lymphocytes (Fig. 2DF) and thrombocytes (Fig. 2DG) were all negative for AKP staining with faint red or faint gray cytoplasm.

### Tetramethylbenzidine reaction for POX

Eosinophils and neutrophils (Figs. 2EA and 2EC) showed positive reactions with coarse, black blue-stained granules in the cytoplasm, the gray blue-stained granules and orange granules were interlaced with each other; monocytes (Fig. 2ED) presented weakly positive with a few blue-stained granules; basophils (Figs. 2EB, 2EE, 2EF and 2EG), large lymphocytes, small lymphocytes and thrombocytes were all negative with light blue cytoplasm.

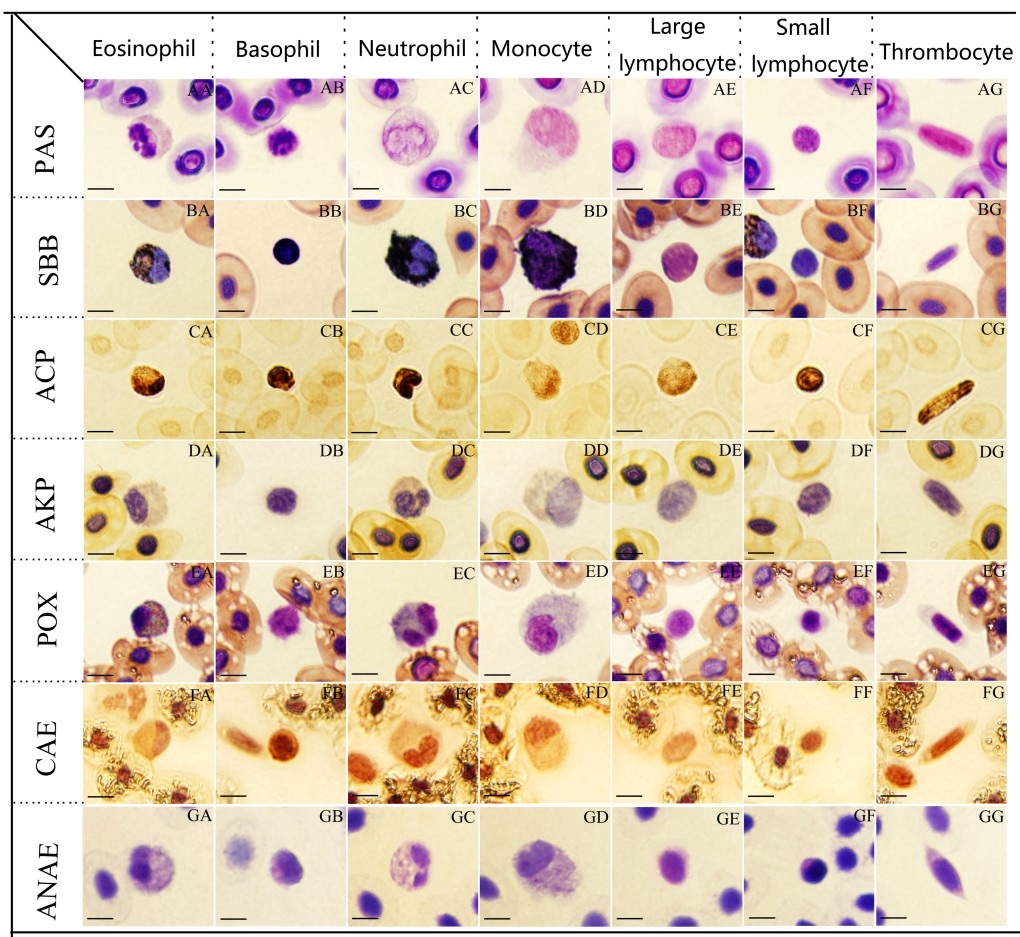

**Figure 2** **The cytochemical staining patterns of tiger frog blood cells.** Eosinophil (AA-GG): (AA) strongly positive reaction with a great number of coarse deep purple red-stained granular deposits, PAS. (BA) positive with numerous dark black-stained granules, SBB. (CA) positive with a large amounts of dark brown granular or diffuse granules, ACP. (DA) weakly positive with light gray diffuse staining in cytoplasm, AKP. (EA) positive with coarse, black blue-stained granules, POX. (FA) positive with a diffusely or granular ruby-colored cytoplasm, CAE. (GA) weakly positive with numerous gray bule-stained granules in the cytoplasm, ANAE. Basophil (AB-GB): (AB) strongly positive for PAS. (BB) positive for SBB, (CB) strongly positive with a great number of black brown granular or diffuse granules, ACP. (DB) negative for AKP. (EB) negative for POX. (FB) positive for CAE. (GB) weakly positive with a diffusely or granular light gray bule-stained cytoplasm, ANAE. Neutrophil (AC-GC): (AC) positive with a diffusely or granular purple red-stained cytoplasm, PAS. (BC, CC, EC) strongly positive with a large number of coarse and dark black-stained granules, which masked the nucleus, SBB. (CC) strongly positive for ACP, (EC) positive for POX, respectively. (DC, GC) weakly positive for AKP and ANAE. (FC) weakly positive with light ruby-colored staining, CAE. Monocyte (AD-GD): (AD, BD,GD) positive for PAS, SBB, ANAE. (CD) weakly positive with a amount of brownish yellow diffuse granules, ACP. (DD) weakly positive for AKP. (ED,FD) weakly positive for POX and CAE. Large lymphocytes (AE-GE): (AE, CE) positive for PAS and ACP. (BE) negative for SBB. (DE, EE, FE) negative for AKP, POX and CAE. (GE) weakly positive for ANAE. Small lymphocytes (AF-GF): (AF,CF) strongly positive for PAS and ACP. (BF) negative for SBB. (DF, EF, FF) negative for AKP, POX, CAE. (GF) weakly positive for ANAE. Thrombocyte (AG-GG): (AG, CG) positive for PAS and ACP. (BG) negative for SBB. (DG, EG, FG) negative for AKP, POX, CAE. (GG) weakly positive for ANAE. (Bar = 10 μm).

**Table 3** The cytochemical staining patterns of leukocytes in tiger frog blood cells.

| Cell types | PAS | SBB | ACP | AKP | POX | AS-D | ANAE |
|---|---|---|---|---|---|---|---|
| Eosinophils | +++ | ++ | ++ | + | ++ | ++ | ++ |
| Basophils | +++ | ++ | +++ | −− | −− | ++ | + |
| Neutrophils | ++ | +++ | +++ | + | ++ | + | + |
| Monocytes | ++ | ++ | + | + | + | + | ++ |
| Large lymphocytes | ++ | −− | ++ | −− | −− | −− | + |
| Small lymphocytes | +++ | −− | +++ | −− | −− | −− | + |
| Thrombocytes | ++ | −− | ++ | −− | −− | −− | + |

**Notes.**
" −" negative; " +" weakly positive; " ++" positive; " +++" strong positive.

### Chloroacetic acid AS-D naphthol staining for CAE

Eosinophils and basophils (Figs. 2FA and 2FB) exhibited positive reactions with a diffusely or granular ruby-colored cytoplasm; neutrophils and monocytes were weakly positive with light ruby-colored staining (Figs. 2FC and 2FD); large lymphocytes, small lymphocytes and thrombocytes (Figs. 2FE, 2FF and 2FG) were negative with light red cytoplasm.

### α-naphthol acetate staining for ANAE

Eosinophils and monocytes (Figs. 2GA and 2GD) presented positive for ANAE staining with numerous gray bule-stained granules in the cytoplasm. Basophils, neutrophils, large lymphocytes, small lymphocytes and thrombocytes (Figs. 2GB, 2GC, 2GE, 2GF and 2GG) were all weakly positive for ANAE staining with a diffusely or granular light gray bule-stained cytoplasm.

## DISCUSSION

### The morphological characteristics of erythrocytes

Erythrocytes play a major role in transporting oxygen and carbon dioxide (*Peng et al., 2018*; *Fathinia et al., 2020*). The mature erythrocytes of tiger frog presented elliptic or long oval in shape with an abundant and faint red-stained cytoplasm free of granules, which were similar to those of non-mammalian vertebrates (*Bricker, Raskin & Densmore, 2012*; *Fang et al., 2014*; *Tang et al., 2015*; *Hernández et al., 2017*). *Fang et al. (2014)* and *Fathinia et al. (2020)* have conferred that the erythrocyte size could reflect the position of a species on the evolutionary scale: the erythrocyte size in lower vertebrates was larger than that in higher vertebrates. However, in our study, the erythrocyte size in tiger frog was smaller than that in some reported reptiles (Table 4), such as European pond turtle (*Emys orbiculuris*), Mediterranean pond turtle (*Muarremys leprosu*) and yellow-bellied slider (*Trachemys scripta scripta*) (*Perpiñán & Sánchez, 2009*; *Hernández et al., 2017*), it was also smaller than that in American bullfrog (*Rana catesbeiana*) and Chinese toad (*Bufo gargarizans*) (*Jin et al., 2015*). *Guo et al. (2002)* have shown that the smaller the erythrocyte, the stronger of its capacity for oxygen transport. Therefore, the phenomenon that erythrocytes in tiger frog were relatively smaller may be related to the stronger activity of this frog. The size of erythrocytes in tiger frogs was not significantly different between sexes, which was inconsistent with that reported in dubois's tree frog whose erythrocytes in females were

**Table 4** The values of RBC, WBC, Neutrophil/lymphocyte ratio and Erythrocyte sizes in tiger frogs and some other species.

| Species | RBC ($10^6$/mm³) | WBC ($10^4$/mm³) | Neutrophil/ lymphocyte ratio | Erythrocyte sizes (μm) | References |
|---|---|---|---|---|---|
| *Rana rugulosa* | 1.12–1.47 | 3.19–4.12 | 0.38–0.41 | 15.09 ± 1.26 × 10.60 ± 0.87 | This study |
| *Polypedates teraiensis* | 0.61–0.65 | 1.15–1.29 | 0.45–0.50 | 19.77 ± 1.53 × 8.63 ± 0.34 | *Madhusmita & Kumari (2014)* |
| *Leptodactylus podicipinus* | 1.40–6.50 | | 0.06–0.34 | | *Franco-Belussi et al. (2021)* |
| *Polypedates maculatus* | 0.40–0.71 | 1.40–2.00 | 0.43–0.45 | 18.43 ± 3.43 × 11.36 ± 1.18 | *Mahapatra et al. (2012)* |
| *Pelophylax bedriagae* | | | | 21.87 ± 1.88 × 12.78 ± 1.44 | *Fathinia et al. (2020)* |
| *Pelodytes caucasicus* | 0.64–0.92 | 0.22–0.35 | | 13.00–17.25 × 8.50–11.25 | *Arikan, Atatür & Tosunoğlu (2003)* |
| *Batrachuperus pinchonii* | 0.072 | 0.2 | 0.09–0.11 | 36.46 ± 0.41 × 21.45 ± 0.26 | *Xiong et al. (2018)* |
| *Trachemys scripta scripta* | | | | 18.42 ± 1.53 × 11.46 ± 0.95 | *Hernández et al. (2017)* |
| *Rana catesbeiana* | | | | 23.21 ± 1.43 × 13.53 ± 1.32 | |
| *Bufo gargarizans* | | | | 19.76 ± 1.17 × 13.71 ± 0.89 | *Jin et al. (2015)* |
| *Emys orbiculuris* | | | | 20.58 ± 1.80 × 12.09 ± 1.27 | |
| *Muarremys leprosu* | | | | 20.53 ± 1.36 × 11.01 ± 0.94 | *Perpiñán & Sánchez (2009)* |
| *Trachemys scripta scripta* | | | | 18.42 ± 1.53 × 11.46 ± 0.95 | *Hernández et al. (2017)* |

significantly larger than in males (*Madhusmita & Kumari, 2014*). The dividing erythrocytes were found sporadically in tiger frog peripheral blood, which is in accord with toad (*Bufo gargarizans*), dubois's tree frog, Chinese sturgeon (*Acipenser sinensis*), sisorid catfish (*Glyptosternum maculatum*), piebald naked carp (*Gymnocypris eckloni*) and prenant's schizothoracin (*Schizothorax prenanti*), (*Guo et al., 2002*; *Gao et al., 2007*; *Zhang et al., 2011*; *Fang et al., 2014*; *Tang et al., 2015*), suggesting that besides the main haematogenic organs, erythrocytes could be produced from amitosis in tiger frog peripheral blood.

## The morphological characteristics of leukocytes

In this study, five types of leukocytes: neutrophils, eosinophils, basophils, monocytes and lymphocytes were observed respectively in tiger frog peripheral blood, which was in accord with those described in neotropical wild frog (*Leptodactylus podicipinus*) (*Franco-Belussi et al., 2021*), American bullfrog and African clawed frog (*Bricker, Raskin & Densmore, 2012*), while most of fish lack basophils (*Fang et al., 2014*; *Zheng et al., 2015*). The morphology of neutrophils, eosinophils and basophils in tiger frogs were usually round, which was consistent with those of other reported amphibians and mammals (*Salakij et al., 2005*; *Prihirunkit et al., 2007*; *Gutierre et al., 2008*; *Bricker, Raskin & Densmore, 2012*). In our study, we found that some neutrophils had a lobed nucleus, usually 2–5 lobes connected by the filaments, which was consistent with that described in caucasus frog, bullfrog, dubois's tree frog and neotropical wild frog (*Arikan, Atatür & Tosunoğlu, 2003*; *Madhusmita & Kumari, 2014*; *Gong et al., 2015*; *Fathinia et al., 2020*). Tiger frog eosinophils were characterized by many coarse, round or rod-shaped, orange acidophilic granules in their cytoplasm, similar to that reported in bullfrog, while different from that of caucasus frog, whose acidophilic granules were usually roundish and stained with bright reddish. In accrodance with that described in caucasus frog, dubois's tree frog and neotropical wild frog (*Arikan, Atatür & Tosunoğlu, 2003*; *Madhusmita & Kumari, 2014*; *Fathinia et al.,*

*2020*), the scarce cytoplasm of tiger frog basophils often contained numerous round and deep purple-stained basophilic granules often masked the nucleus. *Gong et al. (2015)* have reported that there were some tiny pseudopodia on the surface of basophils in bullfrog. However, in this study, we found that there were no pseudopodia on the surface of basophils in tiger frog. Lymphocytes could be divided into large and small lymphocytes, depending on their size. It has been suggested that large lymphocytes and small lymphocytes were different states during their development (*Guo et al., 2002*). Large and small lymphocytes from tiger frogs were morphologically similar to those reported in other amphibians (*Arikan, Atatür & Tosunoğlu, 2003*; *Gutierre et al., 2008*; *Madhusmita & Kumari, 2014*). Monocytes usually present a various and irregular appearance with some vacuoles in the cytoplasm, which was similar to those reported in the American bullfrog and African clawed frog (*Bricker, Raskin & Densmore, 2012*). *Wang, Zhang & Jiang (2001)* have conferred that the vacuoles in monocytes may be related to phagocytosis.

## The morphological characteristics of thrombocytes

The thrombocytes in lower vertebrates were equivalent to mammalian platelets in their function (*Peng et al., 2018*). However, there were great differences in their morphology. Mammalian platelets are enucleated and mostly round to oval disk-shaped (*Salakij et al., 2000*; *Prihirunkit et al., 2007*; *Techangamsuwan et al., 2010*), while thrombocytes in lower vertebrates mostly present round, teardrop, fusiformis or spindle in shape, with a nucleus (*Salakij et al., 2002*; *Chansue et al., 2011*; *Bricker, Raskin & Densmore, 2012*; *Fang et al., 2014*); The thrombocytes of the tiger frog mostly presented long rods or spindle in shape with clear edges and distinct boundaries between nucleoplasms, this morphological characteristics was in accord with that described in American bullfrog (*Bricker, Raskin & Densmore, 2012*) and king cobra (*Ophiophagus hannah*) (*Salakij et al., 2002*), but different from dubois's tree frog (*Madhusmita & Kumari, 2014*), yellow-bellied slider (*Trachemys scripta scripta*) (*Hernández et al., 2017*) and captive bobtail lizard (*Tiliqua rugosa*) (*Moller, Gaál & Mills, 2016*), whose thrombocytes often presented round or oval in shape.

## The blood cell counts and percentage of leukocytes

In this study, the erythrocyte count was more in females than that in males, which was consistent with that reported in Indian rhacophorid tree frog (*Polypedates maculatus*) (*Mahapatra et al., 2012*) and Dubois's Tree Frog (*Polypedates teraiensis*) (*Madhusmita & Kumari, 2014*) *Glomski et al. (1997)* reported that the erythrocyte count of blood varies between 500,000 and 1,500,000/mm$^3$ on average in anurans. The erythrocyte count of tiger frogs were within this range. Female tiger frogs tend to have more leukocytes than males, but the difference was not statistically significant. Similar reasults were reported by *Mahapatra et al. (2012)* in Indian rhacophorid tree frog, whose females showed higher leukocytes count than males, likewise, the difference was not statistically significant.

The percentage of leukocytes were significantly different among different amphibian species. Lymphocytes were the most abundant in American bullfrog, followed by neutrophils and basophils, eosinophils and monocytes were the fewest (*Bricker, Raskin & Densmore, 2012*), while basophils were the most abundant leukocytes of African clawed

frog, followed by lymphocytes and neutrophils, monocytes were significantly fewer, eosinophils were the fewest (*Bricker, Raskin & Densmore, 2012*); lymphocytes were the most abundant leukocytes of dubois's tree frog, followed by neutrophils and eosinophils, monocytes were significantly fewer, basophils were the fewest (*Madhusmita & Kumari, 2014*). In this study, small lymphocytes were the most abundant leukocytes, followed by large lymphocytes and neutrophils, the monocytes and eosinophils were significantly fewer, basophils were the fewest. The neutrophil to lymphocyte (N/L) ratio has been known to reflect levels of stress hormones in vertebrates, and often been used by herpetologists to assess stress levels of amphibians (*Davis & Maerz, 2008*). The reference range of amphibian N/L ratio is between 0.01 to 0.67 (*Davis, 2009*), as described in other frogs (Table 4), the N/L ratio of tiger frog is within this range.

## The cytochemical patterns of different leukocytes

Eosinophils are cells that play a role in phagocytosis and bactericidal effect, and actively participate in the defense against parasitic infections (*Kay, 1985*). In this study, eosinophils were strongly positive for PAS and positive for SBB, POX, ACP, CAE, ANAE, weakly positive for AKP staining, which was different from those described in human whose eosinophils were negative for ACP, AKP and ANAE staining (*Xu et al., 2005*). This cytochemical pattern was also different from that of American bullfrog eosinophils which were positive for SBB, POX, and negative for AKP, CAE staining; and African clawed frog eosinophils which were weakly positive for POX, CAE, and negative for SBB, AKP staining (*Bricker, Raskin & Densmore, 2012*). The strongly positive reaction to PAS and positive to SBB suggested that tiger frog eosinophils contained a mount of glycogen and lipid, which are the important energy source of phagocytosis, moreover, the presence of ACP, AKP, POX, CAE in tiger frog eosinophils indicated that they may play a significant role in phagocytosis and bactericidal effect, but functional tests would be necessary to confirm. Additionally, the positive reaction for ANAE in tiger frog eosinophils indicated a difference from other species.

Human basophils were characterized by numerous coarse, blue-purple and unevenly distributed basophilic granules, which mainly participate in allergic reactions with a relatively weak phagocytic ability (*Falcone, Zillikens & Gibbs, 2006*). In this study, a number of basophilic granules were also found in tiger frog basophils with strongly positive for PAS, ACP, and positive for SBB, CAE, while negative for POX and AKP staining. This cytochemical pattern was different from that of American bullfrog and African clawed frog, whose basophils were negative for SBB staining (*Bricker, Raskin & Densmore, 2012*), while it was generally similar to that described in human, except that the intensity in PAS, SBB and ACP staining of tiger frog basophils was slightly stronger.

Neutrophils are important phagocytes and they are also vitally important in the immune system (*Roos, Voetman & Meerhof, 1983*). In this study, neutrophils were strongly positive for ACP, SBB, and positive for PAS and POX staining, this cytochemical pattern was similar to that described in human, some reptiles and amphibians (*Xu et al., 2005*; *Casal & Orós, 2007*; *Chansue et al., 2011*; *Cooper-Bailey et al., 2011*; *Bricker, Raskin & Densmore, 2012*; *Hernández et al., 2017*). The strongly positive for ACP, SBB, and positive for PAS and POX

staining indicated that tiger frog neutrophils may have a strong capability of phagocytosis and bactericidal effect. A small mount of ANAE was found in tiger frog neutrophils, similar characteristics were also reported in the American bullfrog and toad neutrophils (*Bricker, Raskin & Densmore, 2012*; *Jin et al., 2015*), while human neutrophils were not stained with ANAE, suggesting that amphibian neutrophils have different enzyme content compared with mammals.

Monocytes in human mainly play a dual role of phagocytosis and antigenic processing (*Azevedo & Lunardi, 2003*; *Shigdar, Harford & Ward, 2009*). In this study, monocytes were positive for PAS, SBB and ANAE, and weakly positive for ACP, AKP, POX and CAE staining, which was similar to those described in human except for AKP (negative in human monocytes); while different from those described in American bullfrog and African clawed frog, whose monocytes were positive for CAE, negative for SBB, POX or ACP (*Bricker, Raskin & Densmore, 2012*). The positive reaction to PAS, SBB, ANAE and weakly positive reaction to ACP, AKP, POX, CAE of tiger frog monocytes indicated that they may have different enzyme content to other amphibians.

Lymphocytes belong to agranulocytes and play a significant role in both innate and acquired immunity (*Shigdar, Harford & Ward, 2009*). In this study, large lymphocytes and small lymphocytes were observed, and their cytochemical pattern was generally similar, except that the intensity in PAS and ACP staining of small lymphocytes was slightly stronger than that of large lymphocytes, which may reflect their different immune defenses abilities. The cytochemical pattern of tiger frog lymphocytes was generally similar to that reported in American bullfrog, African clawed frog and toad except for CAE and ANAE staining (*Bricker, Raskin & Densmore, 2012*; *Jin et al., 2015*). The presence of glycogen, ACP and ANAE in tiger frog lymphocytes may suggest functional differences.

### The cytochemical pattern of thrombocytes

Thrombocytes are known to play a significant role in hemostasis and coagulation (*Peng et al., 2018*). In this study, thrombocytes were positive for PAS and ACP, and weakly positive for ANAE, while negative for SBB, POX, AKP and CAE staining. This cytochemical pattern was similar to that described in human platelets except for the PAS staining (*Xu et al., 2005*), while different from that of thrombocytes from toad which were only strongly positive for ACP staining (*Jin et al., 2015*). Some vacuoles have been reported to be present in the cytoplasm of thrombocytes in fish species (*Gao et al., 2007*; *Fang et al., 2014*; *Michał, Beata & Wiesław, 2019*) and it has been considered to be related to the phagocytic function of thrombocytes. In contrast to these fish species, we found no vacuoles in tiger frog thrombocytes. *Wang et al. (2021)* suggested that the positive reaction for PAS and ACP in thrombocytes may reflect their phagocytic ability. Thus, in our study, the thrombocytes of tiger frogs were positive for PAS, ACP, and weakly positive for ANAE staining, indicating that they may have some phagocytic and antigen-presenting functions.

## CONCLUSIONS

In conclusion, this study presented the first comprehensive investigation on the classification, counts, percentage, morphological features and cytochemical patterns of

tiger frog peripheral blood cells. The RBC count of tiger frogs were within the range of the corresponding values in anurans. The percentage of various leukocytes was basically consistent with that reporeted in other amphibians. In addition, the blood cell morphology of tiger frogs was generally similar to that of other amphibians, while their cytochemical patterns had some notable species specificity. Our study could enrich the knowledge of peripheral blood cell morphology and cytochemistry in amphibians, and provide baseline data for health condition evaluation and disease diagnosis of tiger frogs. However, the influences of diseases and environmental factors, such as parasites, herbicides, pesticides and industrial wastewater, on blood cell morphology, hematological parameters and immune functions, remain to be further studied.

## ACKNOWLEDGEMENTS

We would like to thank Yu Xiaoqing for her help in photo processing. We also thank Wu Zhengjie for help with the revision of this manuscript.

### Funding

This work was funded by the key program of the Education Bureau of Anhui Province (Grant No. KJ2013A126) and the Natural Science Foundation of Anhui Province (Grant No. 1808085MC82). The funders had no role in study design, data collection and analysis, decision to publish, or preparation of the manuscript.

### Grant Disclosures

The following grant information was disclosed by the authors:
Education Bureau of Anhui Province: KJ2013A126.
Natural Science Foundation of Anhui Province: 1808085MC82.

### Competing Interests

The authors declare there are no competing interests.

### Author Contributions

- Xianxian Chen conceived and designed the experiments, performed the experiments, analyzed the data, prepared figures and/or tables, authored or reviewed drafts of the article, and approved the final draft.
- Yu Wu performed the experiments, prepared figures and/or tables, authored or reviewed drafts of the article, and approved the final draft.
- Lixin Huang performed the experiments, prepared figures and/or tables, authored or reviewed drafts of the article, and approved the final draft.
- Xue Cao performed the experiments, prepared figures and/or tables, and approved the final draft.
- Misbah Hanif conceived and designed the experiments, prepared figures and/or tables, authored or reviewed drafts of the article, and approved the final draft.

- Fei Peng conceived and designed the experiments, authored or reviewed drafts of the article, and approved the final draft.
- Xiaobing Wu conceived and designed the experiments, authored or reviewed drafts of the article, and approved the final draft.
- Shengzhou Zhang conceived and designed the experiments, analyzed the data, authored or reviewed drafts of the article, and approved the final draft.

## Animal Ethics

The following information was supplied relating to ethical approvals (*i.e.*, approving body and any reference numbers):

This work was approved by the ethics committee of Anhui Normal University (Approval No. 201811). All the handling and sampling were performed in compliance with standard vertebrate protocols and veterinary practices, and in accordance with national and provincial guidelines.

## Data Availability

The raw measurements are available in the Supplemental Files.

## Supplemental Information

Supplemental information for this article can be found online at http://dx.doi.org/10.7717/peerj.13915#supplemental-information.

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
