# Peer review of "Morphology and cytochemical patterns of peripheral blood cells of tiger frog (Rana rugulosa)"

_PeerJ, doi:10.7717/peerj.13915_

## Round 0.1 · original submission · Major Revisions

Thank you for your submission to PeerJ. This article has been reviewed by 3 experts in the field, and all have provided constructive criticism that if addressed will improve the manuscript.

I agree with the reviewers in both 1) yes, this baseline health parameterization is important and very understudied/underreported in the literature, thus it provides important work and 2) the context of the importance of this work has not been clearly or convincingly stated in the abstract nor in the main text. The authors will need to clarify why this work was undertaken, and why should the reader care about it.

In the abstract, please reduce the specific results to a couple of sentences, and provide context for what these parameters might mean or how they can be used.

Note: if there is no statistical difference between groups (male and female) you report the mean and sd for both groups combined - not the mean for the females and the mean for the males - because they are not different from each other.

I'm confused as to why no analysis was conducted across the different staining techniques. Based on the set up in the intro and the descriptions in the methods, as a reader, I am expecting a comparison of staining techniques. If that is not what you intended to do, I request that you restructure your introduction to provide context for what it is you are testing.

Can you compare the shape of each of these white blood cells with what other species of frogs have? Comparing them to known published accounts is fine.

The reviewers mention that the vast proportion of this manuscript is descriptive. and this is fine, but it would benefit from being comparative in some way. The descriptions of the different cell types might be better suited to supplemental materials - it reads almost like a textbook or teaching material for students, not a research article. it is important information and something that should be easily accessible to others who might want to explore this topic. But perhaps it doesn't belong as a main result in this article.

I think the manuscript could use a shift in perspective. What is it that you want your readers to understand? And direct them throughout the article.

PeerJ values itself on the ability to include all data within the article submission. This includes video and photographic data. Please include all photos that were taken and used in this analysis.

·

Basic reporting

The manuscript is well presented, with good language and understandable to readers in the field.
The literature presented had a good selection but only 12.64% are from the last 5 years.
The results are relevant and important for the knowledge of the animal's biology as well as its conservation in the environment.

Experimental design

The experimental design is clear, as are the hypothesis and objectives. There are no mistakes in your presentation.

Validity of the findings

The hematology of wild animals is poorly studied and we lack information in this area. It is important to emphasize the different aspects of amphibian and reptile hematology due to their immediate response to environmental variations.

Additional comments

The beginning of the discussion can be improved with the inclusion of a table or figure with the comparative data from the literature presented, facilitating the understanding of what is intended to be presented.

Reviewer 2 ·

Basic reporting

Abstract

Overall, the abstract includes necessary study steps. However, there are fundamental errors in writing - review with a fluent speaker. The conclusion of the study was not reported.

Introduction

This is a descriptive manuscript emphasizing the morphological characteristics of erythrocytes, leukocytes, and thrombocytes, interesting for possible ecological analyses. However, other aspects that govern hematological dynamics, such as quantitative hematimetric variables (volume, hemoglobin, hematocrit), were not considered in the study. However, the importance of studying blood morphology is not contextualized in the introduction. Likewise, the use and dyeing particularities and relevance are not considered in this topic. Page 90 - 92, the authors cite that ....in recent decades a pandemic loss in amphibian biodiversity has... So, is the Rana rugulosa species in danger or about with the destruction of the environment? If so, how is it widely distributed in much of China and used as human food? Is this species representative of anuran amphibians in the study region? Still, is there an implication between the knowledge of the cytochemistry of blood cells in this species with possible immunological alterations related to the environment? These questions seem crucial to understanding the importance of studying a species that does not seem to be suffering direct effects from the environment.
Materials and methods
Specific numbers for males and females must be explained in the text.
The cell count and morphometric method should be described separately, not in the statistical analysis. In the statistical analysis, consider only the experimental design and the tests used. Here there seems to be an error. ANOVA appears to have been two-way (sex x cell type); explain this model better.
The abbreviations of the dyes/cytochemical reactions must be explicit in the text;

Results

The results are described according to the MM topic.
According to a logical sequence, the cell count should be in Tab. 1, followed by morphometry in Tab. 2, and then the cytochemistry in Tab. 3.
The Tab 1. caption is wrong; it is not about morphology but morphometrical traits. Include the n in each sex and not the total.
The blood count has a different measurement unit about Tab.2 of the discussion topic. Standardize the counting unit.
It is usual and common that both the absolute count and its percentage are included in the same results table, considering the animal blood count. Thus, the authors could consider joining tables 1 and 2, facilitating the discussion and interpretation of the findings.

Discussion

Overall, the discussion is in light of the findings, and the topics are well confronted with the literature. In addition, the subdivision into topics makes it easier to read.
Considering a significant difference between large and small lymphocytes, the authors could have speculated this difference and its significance in this species. This data is relevant (Tab. 2).
The authors cite the morphological characteristics of thrombocytes but do not discuss why this category of cells has a similar prevalence to erythrocytes in the blood and the meaning of this critical finding (Table 2).
Make it clear whether or not thrombocytes have vacuoles. Is this relevant information to this species? Make it clear what is the relevance of positivity to PAS and ACP.

Conclusion

The conclusion should be more precise concerning the results and discussion of the study. The authors report the results between lines 450 - 453 and not their conclusion. In line 454 ...morphology and chemistry in frogs... I do not recommend expanding to other species, given the marked differences. I suggest focusing only on Rana rugulosa.

Experimental design

The experimental design should be revised. The effect of sex and cell type must be considered together (two-way Anova).

Validity of the findings

The study is relevant and well-executed, although it appears to be regionally relevant. Still, there are dubious points that the authors should better explain. In addition, a general revision in the formatting and organization of the introduction, material and methods, results, and discussion statements should be improved.

Additional comments

All comments are on the basic reporting topic.

Reviewer 3 ·

Basic reporting

Most of the article contains clear and unambiguous professional English. It could benefit from additional minor editing. The sentence structures are simple, but this fits the direct, concise reporting of the data. It does read a bit repetitive, particularly concerning the stains and the results. Some of the statements are vague or noncommittal, particularly with use of the words "may" and "could." These will have a better message stated directly with support; I agree that blood cell types may better define health status of frogs, likely at the population level. But how will they do so? Some of the cited sources give evidence for these statements.

The literature references are sufficient, though the authors could include additional sources to support the application of the data they found. Particularly with reference to cell types and their functions.

Overall, the article structure is professional, clear, and easy to follow. The breakdown of sections is straightforward, and makes interpretation easy. Raw data are shared appropriately.

The paper reads more like a report than hypothesis-driven research. The data are important, but may not be best represented in a research article of this nature. Or, the authors can make stronger cases for why the blood cell types share the characteristics they report, and better emphasize how structure impacts function.

Experimental design

The research is primary. However, it is more report than hypothesis-driven. The data are necessary, and often lacking in literature. A research article may not be the best location for this information. But, the article can be adjusted with focus on applicability, with results still reported.

The knowledge gap is evident. The research question is simple, and well-defined. The report-like nature of this paper could fit into a data section alone; the researchers should add more interpretation and future research directions to enhance the message of this work.

The investigation was rigorous, and appropriate for their target data. The methods provide sufficient detail and information for replication, though the value of such work may not be evident.

It is important to describe the frog population, why the selection is valid, and how the addition of other populations of the same species in different macro- and micro-environments may impact overall results.

Validity of the findings

The findings are valid and the data are useful in the field of eco-immunology. However, the hypotheses are vague overall; the authors should provide better focus on how the data can be applied. Otherwise, it should only be reported as a technical note, or as an addendum in other literature.

The underlying data have been provided. The statistical tests are appropriate when three or more variables are compared. Other statistical tests could provide more nuanced interpretation of the data; the assumption that groups are independent may not be valid for all cell types/individuals. This can be especially difficult to maintain when comparing between different populations.

Conclusions are well-stated and limited to supporting results. However, the succinct nature of the results could be better presented with more interpretation and application.

Additional comments

Overall, these data are important to share. As a research article, the message should be adjusted. The article reads too much as a report of data, with little interpretation or direction for future research. The statistical tests are valid, but do not hold as much meaning when interpreting health status. Additionally, the lack of statistical difference may be important from an immunological standpoint, in this species compared to others, or in comparisons between different populations.

I think the images are incredibly valuable, and should make their way into some reference book or literature related to amphibian health/immunology.

---

## Round 0.2 · Major Revisions

Thank you for your resubmission to PeerJ. the authors did a great job addressing the reviewer comments, and the manuscript is nearly ready for publication.

This is an important paper, and we are doing blood cell counts and assessments in a current project I have going and this is going to be a very useful paper – we were just talking about how baseline health data is very much lacking for frogs. Your photos are sure to be incredibly helpful for me and many others as well.

In my reading of the manuscript this time I noticed some potential issues with - or improper use of - statistical analyses. I think that inappropriate statistical tests were conducted, and therefore either need to be explicitly justified or re-analysed. All of your data was analysed with ANOVAs, but it is likely that these are not the most appropriate for all the data collected.

Proportion data typically do not fit the model assumptions for ANOVA – i.e., proportions follow a beta distribution, not a normal distribution. Please double-check that these analyses are correct. Typically, you’ll want to run a GLM with beta distribution for this kind of data. If I correctly remember a beta distribution is not possible in SPSS. Historically folks had analysed proportion data using an arcsin transformation, but current biostats experts believe this is an incorrect use of transformations and therefore should not be completed.

Cell counts would likely follow a Poisson distribution and need to be analysed as a GLM, not ANOVA. However, model assumptions for an ANOVA might be met with a log transformation. It’s unlikely that untransformed data fit model assumptions. Please outline how you determined that model assumptions were met.

Your morphometric data will likely follow a normal distribution but you will need to outline how you determined this. Because you took measurements for multiple cells within a single individuals you’ll need to do a repeated measures analysis: a mixed effects model when individual is a random effect. An ANOVA is not sufficient – unless you took the average across all your cells for each individual, but that wasn’t specified in your methods, nor is it the proper way to perform your stats.


Minor comments:
Frogs have heterophils – not usually labelled neutrophils from what I am aware. Please double-check this and cite as needed.

Any parasites found? In farmed animals, you might expect some blood parasites.

Line 11 p 10 (continuous line counts are more helpful – note that your pages say 1 after page 9) you use acronyms instead of using 1 word to determine the type of cell. I prefer erythrocyte, leukocyte and thrombocyte counts instead of the acronym. But if you need to use the acronym, call them red blood cell, white blood cell etc to match the acronym.

Reviewer 2 ·

Basic reporting

Overall, the current text presents substantial changes from the first version. In all sections, there are profound modifications that have improved the understanding and connections between the hypothesis, results, discussion, and conclusion. Overall, the current text presents substantial changes from the first version. In all sections, there are profound modifications that have improved the understanding and connections between the hypothesis, results, discussion, and conclusion. New and providential information on the importance of the study, methodology, and results has been added. The abstract was changed entirely, and the understanding of the study improved.

Experimental design

It is adjusted to the study, and it is adequate. In addition, corrections to acronyms and distribution of sections were provided.

Validity of the findings

According to the first version, the findings are valid and exciting from a scientific point of view. Now, they are more suitable and have been better organized in the description of the results.

Additional comments

There are no further comments. However, I believe that a deep language review could elucidate specific manuscript points. The key point of the new review is the contextual suitability of the study and its importance, focusing on the Rana regulosa species as a feasible model for hematological references in other species and studies.

---

## Round 0.3 · Minor Revisions

Unfortunately, we are going to have to do one more round of revisions based on the statistical analyses. Great job with the Poisson distribution GLM for the counts, but your morphometric data should still be normally distributed, not follow a Poisson distribution; therefore, using a Poisson distribution will lead to inaccurate statistical results. And furthermore, a gaussian distribution is the same thing as a normal distribution: a GLM with a gaussian distribution is just an LM. Typically for proportions (bounded by 0 and 1) you need to use a beta distribution or quasibinomial distribution GLM. Unless you can justify that the assumptions of a linear model are met, you will have to redo these analyses.

To reiterate, your cell count data appears to have been analysed correctly, but your cell proportion data and cell morphology data is still incorrectly analysed.

It is critical that the assumptions of your LMs or GLMs are met if the analyses are going to be interpreted, therefore you will need to justify that you understand the assumptions, and clarify that they are met.

I see that you added a column in your table for p-value, which is an improvement. But you will need to report all statistical results including the test statistic, degrees of freedom, p-value and model used, as well as what this p-value is comparing. Here your only predictor variable is sex so please make that clear in the table or table legend.

---

## Round 0.4 · accepted · Accept

Thank you for this revised document, Congrats, it is now acceptable for publication.